Contribution of hippocampal BDNF/CREB signaling pathway and gut microbiota to emotional behavior impairment induced by chronic unpredictable mild stress during pregnancy in rats offspring

Zhao Feng 1 2 3 4
Wang Kai 2 3
Wen Yujun 1
Chen Xiaohui 2 3
Liu Hongya 2 3
Qi Faqiu 2 3
Fu Youjuan 2 3
Zhu Jiashu 2 3
Guan Suzhen 1 2 3 guansz_nx2017@sina.com
Liu Zhihong 2 3 lzh_2580@sina.com
1 Ningxia Key Laboratory of Cerebrocranial Disease, Ningxia Medical University , Yinchuan, Ningxia , China
2 Key Laboratory of Environmental Factors and Chronic Disease Control, Ningxia Medical University , Yinchuan, Ningxia , China
3 School of Public Health and Management, Ningxia Medical University , Yinchuan, Ningxia , China
4 School of Public Health, Chongqing Medical University, Chongqing, China, Chongqing Medical University , Chongqing , China
Georgian Badicu
Electronic publication date: 2022 Jun 24
Publication date: 2022
Volume: 10
Electronic Location ID: e13605
Received 2022 Mar 1; Accepted 2022 May 26
Copyright: © 2022 Zhao et al.
Copyright year: 2022
Copyright holder: Zhao et al.
License: This is an open access article distributed under the terms of the Creative Commons Attribution License, which permits unrestricted use, distribution, reproduction and adaptation in any medium and for any purpose provided that it is properly attributed. For attribution, the original author(s), title, publication source (PeerJ) and either DOI or URL of the article must be cited.
License URL: https://creativecommons.org/licenses/by/4.0/

Keywords: Chronic stress, Pregnancy, Gut microbiota, BDNF/TrkB signaling pathway, Offspring, Impairments in emotional behavior

Funding: National Natural Science Foundation of China No 81960591 Key Project of Ningxia Natural Science Foundation No 2022AAC02030 Key scientific research projects of natural science of Ningxia Medical University in 2020 No XZ2020002 This study was supported by National Natural Science Foundation of China (No: 81960591), the Key Project of Ningxia Natural Science Foundation (No: 2022AAC02030), and the key scientific research projects of natural science of Ningxia Medical University in 2020 (No: XZ2020002). The funders had no role in study design, data collection and analysis, decision to publish, or preparation of the manuscript.

==============================
Background

Numerous studies have shown that exposure to prenatal maternal stress (PMS) is associated with various psychopathological outcomes of offspring. The accumulating evidence linking bacteria in the gut and neurons in the brain (the microbiota-gut-brain axis) has been aconsensus; however, there is a lack of research on the involvement mechanism of gut microbiota in the regulation of the BDNF/CREB signaling pathway in the hippocampus of prenatally stressed offspring.

Methods

Pregnant rats were subjected to chronic unpredictable mild stress (CUMS) to establish the prenatal maternal stress model. The body weight was measured and the behavioral changes were recorded. Offspring were tested to determine emotional state using sucrose preference test (SPT), open-field test (OFT) and suspended tail test (STT). Gut microbiota was evaluated by sequencing the microbial 16S rRNA V3–V4 region, and the interactive analysis of bacterial community structure and diversity was carried out. The expression of hippocampal BDNF, TrkB and CREB mRNA and proteins were respectively measured using RT-PCR and Western blotting.

Results

Prenatal maternal stress increased maternal plasma corticosterone levels, slowed maternal weight gain and caused depression-like behaviors (all P < 0.05). In offspring, prenatal maternal stress increased plasma corticosterone levels (P < 0.05) and emotional behavior changes (depression-like state) were observed (P < 0.05). The species abundance, diversity and composition of the offspring’s gut microbiota changed after the maternal stress during pregnancy (P < 0.05). Compared with the control group’s offspring, the species abundance of Lactobacillaceae was dropped, while the abundance of the Muribaculaceae species abundance was risen. Concurrent, changes in the hippocampal structure of the offspring and decreases in expression of BDNF/CREB signaling were noted (P < 0.05).

Conclusions

Prenatal maternal stress leads to high corticosterone status and abnormal emotion behavior of offspring, which may be associated with the abnormal BDNF/CREB signaling in hippocampus of offspring caused by the change of gut microbiota composition.

Introduction

Repeated traumatic events induce long-lasting stress, which affect cognitive, emotional, social behaviors and quality of life in non-specific ways (Barik et al., 2013; Bloomfield et al., 2021; Costa-Mattioli & Walter, 2020). Pregnant woman cannot be separated from the influence of various social and psychological events. When pregnant women face too many stress events and are unable to deal with them, they can be considered to be in a state of prenatal maternal stress (Fleming et al., 2018). The origin of adult health and disease development (DOHaD) hypothesis had pointed out that the adverse intrauterine environment caused by the pathophysiological experience of pregnant woman will change the fetal growth and development planning, and may have a long-term impact on the health of future generations (Codagnone et al., 2019; Hampton, 2004).

The development process is characterized by multiple time windows in which organisms have strong plasticity and are very sensitive to environmental stimuli (Musillo, Berry & Cirulli, 2022). Embryonic stage is the key period of brain development, studies have found that prenatal maternal stress have an adverse effect on the emotional and cognitive development of their offspring (Chagas et al., 2021; Charrier et al., 2022; Glynn & Baram, 2019; Marchisella et al., 2021; Sandman et al., 2018). The hypothalamic pituitary adrenal (HPA) axis is a physiological system and play a key role in regulating the relationship between prenatal maternal stress and offspring development outcomes (Constantinof, Moisiadis & Matthews, 2016; Nakamura, Osonoi & Terauchi, 2010).

Gut microbiota community is a dynamic entity, which composition and activities will change with the changes of host factors and environmental factors in our life (Dahl, Rivero Mendoza & Lambert, 2020; de la Cuesta-Zuluaga et al., 2019; Vich Vila et al., 2020). A growing body of research have revealed that gut microbiota is crucial to the function of our immune system (Dabke, Hendrick & Devkota, 2019) and even the development of various organs (De Vadder et al., 2018). The gut microbiota-brain axis is a two-way communication system between the gastrointestinal and neuroendocrine systems of the body (Morais, Schreiber & Mazmanian, 2021; Zijlmans et al., 2015). Studies have found that prenatal maternal stress induces gut microbiota imbalance, disrupts the neuroimmune network, upregulates the level of placental inflammatory factors, and may increase the susceptibility of offspring to psychiatric disorders through vertical transmission (Antonson et al., 2020; Naudé et al., 2020; Sun et al., 2021). In fact, some clinical studies have shown that prenatal maternal stress is related to the composition of infant fecal microbiota (Aatsinki et al., 2020; Jašarević et al., 2018); at the same time, it is of interest that there is evidence suggesting that altered gut microbiota in mothers and offspring has been linked to cognitive and behavioral abnormalities in adult offspring due to prenatal stress. (Dawson et al., 2021; Vuong et al., 2020). Still, the pathway connections and interactions between the brain and the gut remain unclear.

The hippocampus is part of the brain’s limbic system associated with generating feelings and emotions such as fear, reward, punishment and pleasure, which is also the key part of the brain damaged by stress (Jacobson & Sapolsky, 1991). Brain-derived neurotrophic factor (BDNF) plays an important role in regulating hippocampal neurogenesis, which mainly affects the proliferation and differentiation of hippocampal neurons (Kellner et al., 2014). Previous studies have publicized that damage to the BDNF, tropomyosin receptor kinase B (TrkB), or cyclic adenosine monophosphate responsive element-binding protein (CREB) pathway can result in neurological and behavioral disorders (Ma et al., 2017). Decreased BDNF expression and neurogenic disorders are associated with prenatal maternal stress induced depression-like behaviour of offspring has been reported in several studies (Gur et al., 2017; Kertes et al., 2017; Numakawa, Odaka & Adachi, 2017). It is, however, not known so far if gut microbiota in the regulation of the BDNF/CREB signaling pathway in the hippocampus of prenatally stressed offspring.

Altogether, the present study aimed to investigate the abnormal expression effects of BDNF/CREB signaling pathway in hippocampus caused by the change of gut microbiota in offspring may be related to the emotional behavior impairment of prenatally stressed adult offspring. Therefore, the study simulated prenatal maternal stress by establishing chronic unpredictable mild stress (CUMS) during pregnancy, and detected the changes of gut microbiota structure and composition by 16S rRNA gene sequencing technology, so as to explore the mechanism of gut microbiota involved in hippocampal BDNF/CREB signaling pathway in emotional behavior impairment of offspring caused by prenatal maternal stress.

Materials and Methods

Materials

Experimental animals

Adult healthy Sprague Dawley (SD) rats were from the Experimental Animal Center of Ningxia Medical University, with the certificate number SCXK (Ning) 2015-0001. A total of 16 female rats weighing (200 ± 20) g were randomly divided into a prenatal maternal stress group (stress group) and a control group with eight rats per group. A total of 12 male rats weighing (200 ± 20) g were randomly divided into a control mating group (four rats) and a stress mating group (eight rats). The female rats in the stress group were reared in individual cages, while the control group had two female rats in each cage; three male rats were raised in each cage. All rats were maintained under standard laboratory conditions (12 h light/dark cycle, temperature 21–23 °C, relative humidity 45–65%, and food and water ad libitum) during first week of adaptive feeding. Experimental operation complied with the relevant regulations of the ethics committee of the Animal Experimental Center of Ningxia Medical University (IACUC-NYLAC-2019-089).

Apparatus and reagents

The 131I cortisol radioimmunoassay kit (Beijing North Institute of Biotechnology, Beijing, China); Miseq PE300 sequencing platform (Shanghai Meiji Biomedical Technology Co., Ltd., Shanghai, China); BDNF antibody, TrkB antibody, CREB antibody, β-tubulin antibody and secondary antibody IgG (Solarbio, USA). Protein electrophoresis apparatus and chemical development apparatus (Shanghai Qinxiang Scientific Instrument Co., Ltd., Shanghai, China), fluorescence microplate reader (Bio-Rad, Hercules, CA, USA).

Methods

Chronic unpredictable mild stress (CUMS)

The experimental operation complied with the relevant regulations of the ethics committee of the Animal Experimental Center of Ningxia Medical University (IACUC-NYLAC-2019-089). Descriptions can be found in the ‘Experimental animals’ section. A 21-day CUMS procedure was adopted to establish an prenatal maternal stress model (Guan et al., 2016). The specific stress methods utilised included: ① tail clamping (1 cm from the tip of the tail, 1 min), ② warm water swimming (1 h at 31 °C), ③ behaviour restraint (30 min), ④ water deprivation (24 h), ⑤ crowded environment (24 h, 8 cages, cage tilted 30°), ⑥ shaking (30 min at 1 time·s-1), ⑦ humid environment (24 h at humidity 60–70%), ⑧ food deprivation (24 h), ⑨ heat stress (5 min at 42 °C). One of the nine different stressors was randomly administered each day. ① ② ③ ④ stressors lasted from 10 a.m. the first day to 10 a.m. the next day and other stressors were applied between 10:00 to 12:00 in the morning. The implementation of all stressors was carried out in a separate room, and after the stress is completed, return to the original room. Rats in control group were not exposed to any stress and were given free access to food and water. The whole experimental process was shown in the Fig. 1.

Figure 1 Experimental protocol.

In this study, chronic unpredictable mild stress (CUMS) was used to establish prenatal maternal stress model. The period was 21 days. After the stress was completed, fresh feces were collected from the mother rat. The fresh feces of offspring were collected at 20 days after birth (PND20) for testing, they was weaned at PND21 and the emotional function were determined using Sucrose preference test (SPT), Open-field test (OFT) and tail suspension test (TST) at PND42. The microorganisms in feces of mother and offspring was determined by the 16S rRNA V3–V4 region sequence in Illumina MiSeqPE300 sequencing technology. The offspring’s plasma was collected PND50, the expression of hippocampal BDNF, TrkB and CREB were respectively measured using RT-PCR and Western blotting.

Mating

Initiation of the stress stimuli on the experimentally stressed females was begun 3 days prior to their introduction to a male. Male rats were introduced into the cages so that each stressed rat had one male rat in the cage with them, and unstressed rats received one male per two females. All rats were checked daily for the presence of sperm, which was used to indicate coitus had occurred. The first day of sperm presence was deemed to be gestational day 0. Control group rats were kept together with the male until both female rats had tested positively for sperm or until day 5, when the male rat was removed from the cage. The female rats were separated from the male rats on the day of confirmation of pregnancy. After the assumed pregnancy, the rats in the control group were fed with two animals, while the rats in the stress group were fed separately until the 18th day of pregnancy. The stressful stimuli applied to the experimental group were not suspended during the mating period. After the stress experiment period, all female rats were anesthetized by intraperitoneal injection of chloral hydrate and tissues were collected. Male rats were used for another follow-up experimental study.

The grouping and feeding environment of offspring rats

The day on which the offspring were born was designated as postnatal day 0 (PND 0). The offspring were weaned from their mothers on PND 21. According to the grouping of female rats, each group will have 16 offspring rats (eight male rats and eight female rats) into the follow-up study, which will be divided into control offspring group and stress offspring group. There are four offspring rats in each cage, and female and male rats are housed in separate cages. Offspring rats were weighed and blood was collected from the ocular canthal vein at postnatal day 28 (PND28). Offspring rats were tested for changes in emotional behavior at postnatal day 42 (PND42). On postnatal day 50 (PND50), offspring rats were euthanized by intraperitoneal injection of chloral hydrate. The brain was then dissected, and the hippocampus were separated and quickly frozen using liquid nitrogen. Frozen samples were stored in −80 °C freezer until testing. The offspring rats that were not included were used for other experimental studies of the research group.

Confirmation of the CUMS model during pregnancy

Measurement of maternal plasma corticosterone level

Venous blood samples (1 mL) were collected from all female rats via the intraocular canthal vein on the day before first stress (baseline) and then on the 1st, 7th, 14th, and 21st days post-stress induction, all between 14:00–16:00. As plasma was required to determine plasma cortisol levels, blood samples were centrifuged at 3,000 rpm for 20 min at 4 °C, and the plasma drawn off and stored at −80 °C. The 131I cortisol radioimmunoassay (RIA) kit was used to determine the plasma cortisol level, and the plasma corticosterone concentration required for this study was obtained according to the conversion formula: Ccorticosterone = 50 × Ccortisol (Liu et al., 2004).

Sucrose preference test (SPT)

In the sucrose preference test (Guan et al., 2016), the female rats were trained to adapt to drinking sucrose water before the test. Two water bottles containing 1% sucrose water were placed in each cage for 24 h to allow free drinking. Afterwards, food and drink were withheld from the rats for 24 h. Then, at 10:00 of the next morning, each female rat was given two pre-weighed bottles of water (one bottle 1% sucrose water, one bottle of pure water), after free drinking for 1 h, two bottles were removed and re-weighed. Tests were performed on the day before first stress (baseline) and then on the 1st, 7th, 14th, and 21st days. The measurement indicators included: ① pure water consumption, ② sucrose water consumption, ③ total liquid consumption, ④ 1% sucrose preference percentage = ②/③ × 100%.

Open-field test (OFT)

The bottom surface of the open box (Guan et al., 2016) is composed of 25 squares of equal area, surrounded by black cubes, with a height of 40 cm, and a length and width of 80 cm each. The rats were put into the open box from the central part, and their movement monitored for 3 min. The horizontal movement score is counted by the number of underside blocks the animal passes, and the animal has more than three paws into the square was counted as one point; the vertical movement score was counted by the number of standing uprights, and 1 time was counted as one point. Each rat was measured on the day before first stress (baseline) and then on the 1st, 7th, 14th, and 21st days (11:00–13:00). Clean the open box thoroughly before the next rat was observed.

Determination of emotional behavior changes in offspring

SPT and OFT

The offspring started the SPT and OFT between PND 42 and PND 44 of the study. The specific steps were the same as noted above. Three consecutive measurements were taken, and the average was calculated.

Suspended tail test (STT)

The tail suspension test (Li et al., 2020) was used for depressive symptoms of offspring and performed at PND45. First, the subject was suspended upside-down using a tail suspension experiment box (about 5 cm from the tip of the offspring’s tail), which safely holds the animal approximately 20 cm above the table. Black cardboard was used to limit the animal’s sight on both sides during the experiment. The main index was the “immobility” time of animals within 6 min; Higher immobility scores indicate “disappointment” or depression. Data collected were consisted of immobility time and the number of struggles. The mean values of daily tests over 3 days was calculated.

Collection of hippocampal tissue from offspring

The offspring were anesthetized by intraperitoneal injection of chloral hydrate on PND50. After the brain was stripped, the hippocampus was separated from it. The whole process was carried out on ice. The right hippocampus of the offspring was stored in a cryotube, quickly put into liquid nitrogen for storage, and then moved to −80 °C freezer for subsequent research; each group selected three left hippocampus and fixed with 4% paraformaldehyde, and stored in a refrigerator at 4 °C.

Hematoxylin-Eosin (HE) staining

As noted above, the left hippocampus was fixed with 4% paraformaldehyde for 48 h, dehydrated, embedded in paraffin. Paraffin blocked tissues were then sliced with a thickness of 5 μm using a microtome, the tissue mounted on glass slides and baked at 56 °C for 6 h, and then stained with HE. For the HE staining, paraffin sections were dewaxed with xylene (2 times × 5 min), followed by gradient alcohol dehydration (100% alcohol, 90% alcohol, 80% alcohol, 70% alcohol, 3 min each). Samples were then washed with distilled water for 1 min before hematoxylin stain solution was added for 3 min. The sample was then rinsed with distilled water for 10 s followed by differentiation with hydrochloric acid and ethanol for 10 s and a rinse with distilled water for 3 min. Eosin stain was then added for 3 min, followed by a distilled water rinse for 5 s, then dehydration using 70% alcohol for 30 s, 80% alcohol for 30 s, 90% alcohol for 1 min, 95% alcohol two times for 3 min each time, 100% alcohol 2 times for 3 min each time. Xylene was then applied two times for 5 min each time, and finally tissue was sealed with neutral gum. The morphological and structural changes in the hippocampal CA1 area were observed under optical microscope.

Quantitative real-time RT-PCR

According to the instructions of the “RNAsimple Total RNA” kit (Tiangen Biotech, Beijing, China). In short, added 700 µl of lysate per 50 mg of hippocampus for homogenization; left it at room temperature for 5 min to completely separate the nucleic acid-protein complexes, added 200 µl of chloroform, and centrifuged at 12,000 rpm for 10 min at 4 °C to obtain the aqueous phase extraction, added 0.5 times absolute ethanol to mix, and centrifuged at 4 °C and 12,000 rpm for 30 s, added 500 μl protein solution, and centrifuged at 4 °C and 12,000 rpm for 30 s, added 500 μl rinse solution, and incubated at 4 °C after centrifugation at 12,000 rpm for 30 s, the RNA pellet was finally dried, and the pellet was dissolved in 50 mL of DEPC water. Extracted RNA was quantified using a nucleic acid protein quantitative instrument. Samples with an A 260/280 ratio of between 1.8–2.0 were used. A three-step reverse-transcription quantitative PCR (RT-qPCR) assay was performed. Reverse transcription (RT) and quantitative polymerase chain reaction (qPCR) were performed in the CFX96 real-time fluorescent quantitative PCR instrument (Bio-Rad, Hercules, CA, USA). BDNF, TrkB and CREB RT-PCR primers used were as follows: BDNF: forward, 5′-TGTGGTCAGTGGCTGGCTCTC-3′, reverse, 5′-ACAGGACGGAAACAGAACGAACAG-3′

TrkB: forward, 5′-GGTCTATGCCGTGGTGGTGATTG-3′, reverse,5′-ATGTCTCGCCAACTTGAGCAGAAG-3′

CREB: forward,5′-GGAGCAGACAACCAGCAGAGTG-3′, reverse, 5′-GGCATGGATACCTGGGCTAATGTG-3′;

β-Actin: forward,5′-TGTCACCAACTGGGACGATA-3′, reverse, 5′-GGGGTGTTGAAGGTCTCAAA-3′.

The specific conditions for the PCR reaction were as follows: pre-denaturation 15 min at 95 °C, denaturation 10 s at 95 °C, annealing 20 s at 55 °C, and extension 30 s at 70 °C, for a total of 40 cycles. The 2−ΔΔCt method was used to calculate the relative expression level of the target gene.

Western blotting

Extract the total protein of hippocampus tissue

To determine the total protein of the hippocampal tissue, 700 µL of cold lysis buffer was added to each 50 mg of hippocampus and homogenized. The homogenate was then centrifuged at 12,000g for 5 min, and the supernatant saved for analysis. Protein concentrations were determined using a BCA assay.

Bicinchoninic acid (BCA) method to measure protein concentration

The instructions provided with the kit were followed carefully: The standard curve was drawn, and an appropriate amount of BCA working solution for reagent A and reagent B (50:1) was prepared. Reagents A and B were thoroughly mixed, and the samples diluted to a suitable concentration. For this experiment, the total volume is 20 L, so 200 µL BCA working solution was used. The solution was then added to each well and mixed thoroughly. The plate was sealed and placed in a 37 °C incubator for 30 min, after which the absorbance at 562 nm was determined using a microplate reader. Sample protein concentration was calculated according to the standard curve.

The specific steps of western blotting

A protein loading buffer of 5×SDS-PAGE was added to the sample at a ratio of 4:1. After denaturing at 100 °C for 5 min, the sample was stored in a −20 °C freezer until use. At the time of use, protein extracts (50 μg) were electrophoresed in 30% SDS-polyacrylamide gels and transferred to polyvinylidene difluoride (PVDF) membranes. Blots were blocked in PBST buffer (2 L ddH2O and 2 mL of 0.05%Tween) with 5% dry milk and incubated with an anti-BDNF antibody (Abcam Biotechnology; 1:2,000), TrkB antibody (Cell Signaling Technology; 1:1,000), CREB antibody (CST; 1:1,000), and β-tublin antibody (Multisciences (Lianke) Biotech Co., Ltd; 1:1,000), and then incubated overnight at 4 °C. Blots were then rinsed 3 times with 1×PBST, for 10 min/time. The secondary antibody, Pierce Goat Anti-Rabbit IgG (Santa Cruz Biotechnology; 1:2,000), was added, and the blot incubated for 1 h at room temperature on a decolorizing shaker. It was then rinsed 3 times with 1×PBST, for 10 min/time. The colour developing solution A and B was mixed, and 1 mL added to the membrane. A ChemiScope 3,000 chemiluminescence instrument was used to detect and take pictures. The integral optical density value of the target protein was calculated and then compared with β-tubulin expression to give the degree of expression of the target protein.

16S rRNA gene sequencing

The fresh feces of the mother rats were collected the next day after the model was built, six pellets per group; the offspring were collected on the 20th day after birth, six pellets per group, all feces were sterile and cryopreserved in a refrigerator at −80 °C. The 16S rRNA gene sequencing technology was used to analyze the difference in the gut microbial communities. Total DNA was extracted using the E.Z.N.A. Stool DNA kit (Omega Bio-Tek, Norcross, GA, USA). The V3–V4 region of the bacterial 16S rRNA gene was polymerase chain reaction (PCR)-amplified using the following primers: 338F 5′-ACTCCTACGGGAGGCAGCA-3′ and 806R 5′-GGACTACHVGGGTWTCT AAT-3′. PCR was performed under the following conditions: 95 °C for 3 min, followed by 27 cycles 30 s each at 95 °C, 55 °C for 30 s, and 72 °C for 45 s, with a final 10 min step at 72 °C. Amplification was confirmed by 2% agarose gel electrophoresis. PCR products were purified with the AxyPrep DNA kit (AXYGEN, Tewksbury, MA, USA) and purified amplicons were pooled in equimolar and paired-end sequenced (2 × 300) on an Illumina MiSeq platform (Illumina, San Diego, CA, USA) according to the standard protocols by Majorbio Bio-Pharm Technology Co. Ltd. (Shanghai, China). The raw reads were deposited into the NCBI Sequence Read Archive (SRA) database (accession number: PRJNA721070). After distinguishing the samples, community bar plot analysis was performed at the phylum and genus levels. The Phylogenetic Investigation of Communities by Reconstruction of Unobserved States (PICRUSt) was used to predict function and obtain to Operational Taxonomic Units (OTU), and to obtain a functional abundance spectrum. The data were analyzed on the free online Majorbio I-Sanger Cloud Platform (Shanghai Majorbio Biopharm Technology Company, Shanghai, China).

Statistical analysis

All data are statistically analyzed used to the Statistical Package for the Social Sciences (SPSS) 23.0, expressed as mean ± SD. The plasma corticosterone level, OFT and SPT of female rats were analyzed by two-way repeated measurement data analysis of variance, the comparison between the two groups at different time points was analyzed using Student’s t test. Bivariate correlations were performed using Spearman co-relations, and P < 0.05 was considered statistically significant. All graphs were constructed in GraphPad Prism 5.0.

Results

CUMS increases maternal plasma corticosterone levels

The analysis of variance for the repeated plasma corticosterone measurements showed that CUMS had a significant impact on the corticosterone level of maternal females (F = 7.717, P = 0.024). The corticosterone level of the stressed group drastically changed with stress (F = 6.076, P = 0.03). T-tests revealed that the plasma corticosterone level of the stressed group was higher than that of the control group following exposure to stress for 7 days (t = 2.341, P = 0.047). At the same time, the plasma corticosterone level of the stress group rose to a peak value which was higher than that of the control group following exposure to stress for 14 days (t = 5.414, P = 0.001), indicating that the experimental group was experiencing a stressful state (Fig. 2A).

Figure 2 Effect of CUMS model on mantel.

(A) Effect of CUMS model on mantel of plasma corticosterone levels; (B) effect of CUMS model on body weight. Data are expressed as the mean ± SD (n = 8 per group); an asterisk (*) indicates comparison with the control group at the same time point, P < 0.05.

CUMS slowed down maternal weight gain

The analysis of variance for the repeated weight measurements found that there was a difference in the body weight of the two groups (F = 4.855, P = 0.032), indicating that stress had an effect on the weight gain. Comparison within the two groups found that the time factor was statistically significant (F = 160.630, P = 0.001), the body weight of rats changed greatly with the increase of stress time; there was an interactive relationship between time and body weight (F = 14.705, P = 0.001), indicating that the role of the time factor varies with the group different. The comparison of different stress time points found that there were differences in the body weight of the two groups of female rats on the 7, 14 and 21 days of stress (t1 = 3.163, P = 0.036; t2 = 4.249, P = 0.024 t3 = 3.752, P = 0.035) (Fig. 2B).

CUMS decreased maternal liquid consumption

The repeated measurement of liquid consumption analysis of variance revealed that CUMS significantly negatively affected sugar water intake, total liquid consumption and 1% sucrose preference in the pregnant rats (F1 = 18.100, P = 0.002; F2 = 9.007, P = 0.013; F3 = 5.528, P = 0.041). Following exposure to stress for 14 and 21 days, it was demonstrated by t test that sucrose-intake of the stress rats was reduced compared with control rats (t1 = 2.557, P = 0.029; t2 = 2.897, P = 0.016) (Figs. 3A–3D).

Figure 3 (A–D) SPT; (E and F) OFT.

Data are expressed as the mean ± standard deviation (SD) (n = 8 per group); an asterisk (*) indicates comparison with the control group at the same time point, P < 0.05.

CUMS reduced maternal horizontal and vertical movements

The repeated measurement analysis of variance revealed that CUMS significantly affected the horizontal and vertical movements in the group (F1 = 36.339, P = 0.001; F2 = 5.680, P = 0.049). After 14 days’ exposure to stress, vertical movements in the stress group were fewer than those in the control group (t = 3.527, P = 0.005) (Fig. 3E), and after exposure to stress for 7 days, horizontal movements in the stress group were fewer than those in the control group (t = 2.950, P = 0.013) (Fig. 3F), predicting CUMS caused a reduction of their activity and less curiosity about the new environment of fetal rats.

Body weight and plasma corticosterone level of offspring changed after maternal stress during pregnancy

Compared with the control offspring, the body weight of the stress offspring at PND28 was lower (t = 2.125, P = 0.042) (Fig. 4A) and the plasma corticosterone was higher of the control group offspring (t = −2.640, P = 0.017) (Fig. 4B), indicating that maternal stress during pregnancy affected the weight gain and stress level of offspring as seen by plasma corticosterone.

Figure 4 Effect of CUMS model on offspring.

(A) PDN28 plasma corticosterone level; (B) PND28 Body weight (n = 16 per group); (C–F) SPT; (G and H) OFT; (I and J) STT (n = 8 per group). Data are expressed as the mean ± standard deviation (SD); an asterisk (*) indicates comparison with the control offspring group, P < 0.05.

Fluid consumption of offspring reduced after maternal stress during pregnancy

Student’s t-test results showed that the control group offspring consumed more sucrose water (t = 3.045, P = 0.01) (Fig. 4D) and 1% sugar percentage (t = 3.804, P = 0.003) (Fig. 4F) than the stressed group offspring, indicating that maternal stress during pregnancy may cause offspring experience a decreased ability to respond appropriately to hedonic events.

Horizontal and vertical movement of offspring descended after maternal stress during pregnancy

The Student’s t test results showed that the vertical and horizontal movement scores of the offspring of the control group were significantly higher than those of the offspring of the stress group (t1 = 2.389, P = 0.032; t2 = 2.545, P = 0.023) (Figs. 4G and 4H), indicating that maternal stress during pregnancy may lead to offspring activity levels are lowered and curiosity about the external environment decreased.

The immobility time of offspring increases after maternal stress during pregnancy

The Student’s t test results showed that the control group offspring spent less time being immobile (t = −3.753, P = 0.005) (Fig. 4I) and struggled more than the stressed group offspring (t = 2.781, P = 0.021) (Fig. 4J). This results suggested that offspring had prolonged intermittent immobility responses after stress during pregnancy, indicating a depression-like state.

CUMS changes the maternal gut microbiota

CUMS reduces the species abundance and diversity of maternal gut microbiota

The Student’s t-test results showed that the Sobs, Shannon, Ace and Chao indexes of the control group were higher than those of the stress group (t1 = 3.419, P = 0.027; t2 = 3.900, P = 0.018; t3 = 3.003, P = 0.040; t4 = 4.139, P = 0.014) (Figs. 5A–5C, 5E), while the control group’s Simpson index was lower than that of the stressed group (t = −5.045, P = 0.007) (Fig. 5D). The results showed that the species richness and diversity of maternal gut microbiota decreased significantly after CUMS.

Figure 5 α diversity analysis of gut microbiota in mantel and offspring.

(A) OTU number of gut microbiota; (B and C) abundance of gut microbiota; (D and E) diversity of gut microbiota; (F) coverage of gut microbiota. Data are expressed as the mean ± SD (n = 3 per group); an asterisk (*) indicates comparison with the control group, P < 0.05. (G) OTU number of gut microbiota in offspring; (H and I) abundance of gut microbiota in offspring; (J and K) Diversity of gut microbiota in offspring; (M) coverage of gut microbiota in offspring. Data are expressed as the mean ± SD (n = 6 per group); an asterisk (*) indicates comparison with the control offspring group, P < 0.05.

CUMS changes the species composition of maternal gut microbiota

The analysis of the species composition of the maternal gut microbiota at family level showed that the predominant gut microbiota were respectively Muribaculaceae, Ruminococcaceae, Prevotellaceae, Rikenellaceae and Lachnospiraceae (Fig. 6A). The species abundance of Ruminococcaceae in the control group (32.67%) was higher than the stress group (23.45%) (Fig. 6C). While the species abundance of Prevotellaceae in the stress group was higher than the control group (P < 0.05) (Fig. 6B). The analysis of the species composition of the maternal gut microbiota at genus level showed that there are differences in the species abundance of gut microbiota between the control group and the stress group (P1 = 0.027, P2 = 0.004, P3 = 0.002, P4 = 0.032) (Figs. 6E–6I) indicating that CUMS changed the species composition of maternal gut microbiota.

Figure 6 Family and genus are levels analysis of gut microbiota in Effect of CUMS model on maternal.

(A) Circos diagram, the left half circle represents the species composition in the sample, the color of the outer ribbon represents the group, the color of the inner ribbon represents the species, and the length represents the relative abundance of the species in the corresponding sample; the right half circle represents the The distribution ratio of species in different samples at the family level. The outer ribbon represents the species, the inner ribbon color represents different groups, and the length represents the distribution ratio of the sample in a certain species; (B) different species of mothers at the family level, the ordinate is the name of the group, and the abscissa is the proportion of species in the group; (C) proportion of sequences in Ruminococcacea; (D) proportion of sequences in Muribaculaceae; (E) different species of mothers at the genus level, the ordinate is the name of the group, and the abscissa is the proportion of species in the group; (F) Proportion of sequences in Prevotellaceae_UCG-001; (G) proportion of sequences in Butyricicoccus; (H) proportion of sequences in unclassified_f__Muribaculaceae; (I) proportion of sequences in Anaerotruncus. Data are expressed as the mean ± SD (n = 3 per group); an asterisk (*) indicates comparison with the control group, P < 0.05; ** P < 0.01.

Changes in the offspring’s gut microbiota after maternal stress during pregnancy

The species abundance and diversity of the offspring’s gut microbiota decreased after the maternal stress during pregnancy

The Student’s t-test results showed that the Sobs, Simpson, Ace, and Chao indexes of the control group offspring were higher than those of the stress group offspring (t1 = 2.505, P = 0.037; t2 = 2.756, P = 0.025; t3 = 5.670, P = 0.000; t4 = 4.748, P = 0.001) (Figs. 5G, 5I–5K). There was no difference between the two groups’ convergence index (t = 1.677, P = 0.132) (Fig. 5M), indicating that the species richness and diversity of offspring’s gut microbiota were significantly reduced after maternal stress during pregnancy.

The species composition of the offspring’s gut microbiota changed at the family level after the maternal stress during pregnancy

The Venn plot (Fig. 7A) was used to reflect the number of species common and unique to the operational taxonomic units (OTUs) levels of the gut microbiota of each group of rats, the number of OTUs unique of the gut microbiota in control offspring was 83 types, higher than 75 types in stress offspring, and the number of OTUs common to the two groups was 582 types. At the genus level, the species with relatively higher abundance are Lactobacillus, norank_f__Muribaculaceae, etc (Fig. 7B). The species abundance of Muribaculum in the control group offspring is higher than that of the stress group offspring (P = 0.043) (Figs. 7C and 7D); The species abundance of Lachnospiraceae_NK4B4_group in the control group’s offspring was lower than that of the stress group offspring (P = 0.045) (Figs. 7C and 7E). These results indicated that maternal stress during pregnancy may cause changes in the species composition of offspring gut microbiota.

Figure 7 Gut microbiota in offspring.

(A) Venn chart of gut microbiota in offspring, different colors represent different groups, the number in the overlapping part represents the number of species shared in multiple groups, and the number in the non-overlapping part represents the number of species unique to the corresponding group; (B) genus level analysis of gut microbiota in offspring, the ordinate is the name of the group, and the abscissa is the proportion of species in the group; (C) different species of offspring at the genus level, the ordinate is the name of the group, and the abscissa is the proportion of species in the group; (D) proportion of sequences in Muribaculum; (E) proportion of sequences in Lachnospiraceae_NK4B4_group. Data are expressed as the mean ± SD (n = 6 per group); an asterisk (*) indicates comparison with the control offspring group, P < 0.05.

Hippocampal structure of offspring damaged due to maternal stress during pregnancy

After H&E staining, the results showed that compared with the control offspring group, the structure of each layer of the hippocampal structure in the stress offspring group was not obvious, the intercellular space was increased and the arrangement was loose, and the cell morphology was irregular (Figs. 8A and 8B), indicating that maternal stress during pregnancy had been changed in the structure of the CA1 region of the hippocampus of the offspring.

Figure 8 Effects of CUMS on the nervous system of offspring histological section of hippocampus.

(A and B) Representative photomicrographs of the hippocampal CAI area (X200); maternal stress during pregnancy reduces the expression of BDNF/CREB signalling pathway related proteins and mRNA in the hippocampus of stressed offspring. (C) WB strip chart, (D) BDNF expression level; (E) TrkB expression level; (F) CREB expression level; (G) BDNF mRNA expression level; (H) TrkB mRNA expression level; (I) CREB mRNA expression level. Data are expressed as the mean ± SD; *compared with the control offspring group, P < 0.05.

BDNF/CREB signaling pathway of hippocampus changed in offspring after maternal stress during pregnancy

Student’s t-test results showed that the expression level of BDNF protein in the stressed group offspring was significantly reduced (t = 2.937, P = 0.041) (Figs. 8C and 8D). At the same time, the expression levels of TrkB and CREB proteins in the stress group offspring decreased (t1 = 3.203, P = 0.032; t2 = 8.259, P = 0.001) (Figs. 8C, 8E and 8F), indicating that the expression of BDNF/CREB signaling pathway-related proteins in the hippocampus of offspring were reduced after maternal stress during pregnancy. Student’s t-test results showed that the expressions of BDNF, TrkB and CREB mRNAs in the stress group offspring were significantly lower than those of the control group (t1 = 11.157, P = 0.000; t2 = 22.446,P = 0.000; t3 = 2.936, P = 0.043) (Figs. 8G–8I), suggesting that hippocampal mRNA expression of BDNF/CREB signaling pathway proteins were decreased in offspring after maternal stress during pregnancy.

Correlation analysis between the genus level of gut microbiota and environmental factors

Correlation analysis showed that 1% sucrose preference percentage was positively correlated with Escherichia-Shigella (P = 0.045), negatively correlated with Oscillibacter (P = 0.003), and negatively correlated with Ruminococcaceae_UCG-010 (P = 0.049), but negatively correlated with Ruminiclostridium (P = 0.024) and Ruminiclostridium_5 (P = 0.001). Immobility time was negatively correlated with Muribaculum (P = 0.011) and Romboutsia (P = 0.032) while body weight was negatively correlated with Muribaculum (P = 0.042). However, Muribaculum was positively correlated with horizontal movement (P = 0.005) and plasma corticosterone (P = 0.034). Plasma corticosterone itself was positively correlated with (P = 0.024) (Fig. 9). These results indicated that there may be a correlation between emotion behavior damage in offspring and gut microbiota.

Figure 9 Correlation analysis between environmental factors and offspring gut microbiota species.

The abscissa represents the environmental factor, the ordinate represents the species name, the legend on the right is the color interval of different R values; * P < 0.05; ** P < 0.01; *** P < 0.001.

Discussion

Offspring born from prenatal maternal stress showed suboptimal neurobehavioral function and hyperactive cortisol responses (Molenaar et al., 2020; Oberlander et al., 2008). In the study, we found that prenatal maternal stress not only increased plasma corticosterone, but also decreased the weight gain of offspring. It has also been discovered that the presence of generalized anxiety disorder during pregnancy increases the incidence of low birth weight infants (Gelaye et al., 2020). Further, using multiple behavioral experiments, the results showed that prenatal maternal stress leaded to the decline of independent and inquiry behavior ability in new and different environment of open-field test, decreased responsiveness to rewards in sucrose preference test and the generation of despair in suspended tail test of offspring, betoken that the prenatally stressed offspring had emergence of emotional behavior damage. These findings are in concordance with previous work which also found stress during pregnancy caused prolonged ‘immobility’ time and depression-like behavior in offspring (Butkevich et al., 2011).

The HPA axis and the gut microbiome display bidirectional communication such that alterations in one system may affect the function of the other (Cryan & O’Mahony, 2011; Cryan et al., 2019; Cusick, DuVal & Cox, 2021). In the study, we investigated the interactive effects of maternal stress and manipulations of the maternal microbiome on offspring growth, gut microbiome composition and diversity, stress response, and emotional behavior.

Prenatal maternal stress changed the species composition and diversity of maternal gut microbiota

In response to challenges such as infection, altered diet and stress during pregnancy, ‘Dysbiosis’ of the maternal gut microbiome occurs (Vuong et al., 2020). At the family level, Ruminococcaceae, Muribaculaceae, Lachnospiraceae, Prevotellaceae, and Rikenellaceae were the dominant bacteria. The current study found that the species abundance of Ruminococcaceae in the stress group (23.45%) was lower (32.67%), while the species abundance of Prevotellaceae in the stress group was higher than the control group. Prevotella was positively correlated with immune traits, and stress would cause it to increase significantly (Garrett, 2020). At the genus level, the species composition of the maternal gut microbiota were different between the control group and the stress group. However, Jiang et al. (2015) found that the Shannon index of alpha diversity in depressive-like individuals was increased, which is contrary to our findings. These variations suggest that prenatal maternal stress may reduce species richness, change the species composition of maternal intestinal microbiome at the family level, and eventually lead to the disorder of maternal intestinal microbiome. However, more studies are needed to verify the results.

Changes in the offspring’s gut microbiota after prenatal maternal stress

It has been shown in mice that the mother’s intestinal bacteria and their metabolites are key to the healthy development of the fetal brain (Vuong et al., 2020) and manipulation of the maternal microbiome or maternal stress can independently impact offspring’s foundation and development of offspring’s microbiome (Dominguez-Bello et al., 2010; Golubeva et al., 2015; Jašarević et al., 2017) and emotion behaviour. Mounting evidence indicates that there is bidirectional communication between the HPA axis and the gut microbiome. The current thinking within the scientific community is that the maternal environment during pregnancy affects the microbiota of foetus before birth, perhaps even helping the foetus to develop proper immune responses (Seferovic et al., 2019; Younge et al., 2019). The study found that pregnancy stress resulted in reduced species richness and diversity of the offspring gut microbiome, while altering the species composition of the offspring gut microbiome at the genus level. This was mainly reflected in the varying abundance of the dominant species, Lactobacillus, which the stressed group’s offspring showed significantly lower levels of this bacteria than that of the control group’s offspring. At the same time, the species abundance of Prevotellaceae in the stressed group’s offspring was lower than that of the control group’s offspring. It is worth noting that previous studies had shown that an abundance of Prevotella was considered a potential characteristic parameter for major depressive disorder (Gu et al., 2020). Along with our study, Golubeva et al. (2015) found that prenatal stress in pregnant mice caused a decrease in the number of lactic acid bacteria in the offspring’s intestines, while showing an increase in the numbers of Oscillatoria and Peptococcus. Zijlmans et al. (2015) found that stress stimulation in late pregnancy can reduce the abundance of Lactobacilli and Bifidobacteria in the offspring’s gut microbiome. Our study showed there was a negative correlation between 1% sucrose preference percentage and the presence of Ruminococcaceae family bacteria and Oscillibacter, while Muribaculum was positively correlated with horizontal movement and immobility time, which indicates that there is a specific correlation between the emotional impairment of the prenatally stressed offspring and the makeup of the gut microbiome.

Changes in the BDNF/CREB signaling pathway in the hippocampus of offspring after prenatal maternal stress

Experiences during prenatal maternal stress (Duckworth, Belloni & Anderson, 2015; Seckl & Meaney, 2004) and exposure to different maternal microbiomes (Jašarević et al., 2017) are important sources of organismal variation that can have long-term effects on offspring. Long-term potentiation of synaptic transmission in the hippocampus is the cellular basis of emotion in vertebrates (Illouz, Madar & Okun, 2020) and the hippocampal CA1 area is a suitable location to look for learning-specific changes in excitatory postsynaptic potential (Shu et al., 2019; Tyrtyshnaia et al., 2020). Functional changes in hippocampal CA1–CA3 synapses and morphological modifications in CA3 pyramidal neurons have been reported to be associated with the occurrence of depression (Qiao et al., 2014). Meanwhile, stress can cause hippocampal neurons to change, shrinking the dendrites of pyramidal neurons, which could damage the memory (Carvalho-Netto et al., 2011). Our findings support the initial hypothesis that hippocampal structure changed in stress group offspring.

BDNF is a major neurotrophic factor that functions in the maintenance and the survival of neurons (Sepehr et al., 2020), TrkB and CREB plays a vital role in depression and antidepressant responses (Sonoyama et al., 2020). Our study found that the expression of BDNF, TrkB and CREB in the offspring’s hippocampus are reduced after maternal stress during pregnancy. It is becoming increasingly evident that CREB has multiple roles indifferent brain regions under various conditions (Mu et al., 2022), and which is not only activated but also upregulated by chronic antidepressants treatment, in particular in the hippocampus (Peter Gass & Riva Marco, 2007). These findings suggest that early maternal stress accelerates infant neurodevelopment in a manner that may increase risk for behavioral problems (Borchers et al., 2021). Interestingly, there is some evidence that prenatal stress exerts a sex-specific effect. For example, prenatal stress resulted in a significant increase of BDNF expression in male offspring, but no effect in female offspring (Zuena et al., 2008). Also, female usually display less anxiety-like behaviors than males (Zagron & Weinstock, 2006). Moreover, male offspring were more sensitive to the prenatal stress in cognition (Shang et al., 2019). The findings suggest that, for females, elevated maternal distress alters fetal development, with implications for postnatal function (Howland et al., 2020). It is unfortunate that we did not take into account the sex-specific effects of offspring emotional behaviour when designing the study, which will be further discussed in our research.

BDNF/CREB signaling pathway in hippocampus of offspring caused by the change of gut microbiota composition after prenatal maternal stress

To continue, the evidence was demonstrating of the fact that gut microbiota plays a noteworthy role in the development and functioning of the CNS (Hosseinifard et al., 2019; Yarandi et al., 2016). Borrelli et al. (2016) indicated that L. rhamnosus administration led to increased brain expression levels of BDNF as well as raised fecal Firmicutes and reduced Proteobacteria population in zebrafish. Correspondingly, Morshedi, Saghafi-Asl & Hosseinifard (2020) illustrated in their study that Synbiotic administration was associated with expressive shifts including BDNF, TrkB, and CREB in the hippocampus, while Lactobacillus/Clostridium and Lactobacillus/Bacteroides singlehandedly improved. Qi et al. (2017) illustrated that tea polyphenols had protective effects against oxidative stress-triggered cognitive impairment via modulation of the BDNF/CREB signaling pathway in housing mice in constant darkness. In the present study, it is to be pointed out that the species with relatively higher abundance are Lactobacillus in prenatally stressed offspring. Along with our study, it is newsworthy to state that Lactobacillus casei can improve the composition of the gut microbiota, and then cause the change of BDNF/CREB signaling pathway, so as to restore the depressive-like behaviour of rats induced by maternal prenatal stress (Gu et al., 2020). From all the research above, it was illustrated that there was a strong correlation between the levels of BDNF/ CREB signaling pathway and the population of Lactobacillus bacteria. On the other hand, if these probiotics bacteria are reduced to zero, the impairment of emotion is certain. Then again, studies in this field are still limited and there is no clear mechanism for the gut–brain interaction in improving emotional behaviour impairment of prenatally stressed offspring.

Conclusions

Prenatal maternal stress can alter the development of offspring and have long-lasting effects on offspring emotional behaviour, that consideration of how the maternal microbiome interacts with prenatal maternal stress and their long-term effects on behaviour development of offspring is still needed. Here, we showed the mother were in a high corticosterone state during prenatal maternal stress, which leaded to gut microbiome diversity and composition disorder of mother and offspring. What is more, we demonstrated that maternal stress may interact with impairment of offspring’s emotional behaviour, which may be associated with the abnormal BDNF/CREB signaling in hippocampus of offspring caused by the change of gut microbiota composition. As a final point, manipulating the gut microbiota can be considered as a therapeutic target for the prevention and treatment of neuropsychological disorders due to prenatal maternal stress. Owing to the highly limited researches in this field, further studies are warranted.

Supplemental Information

Supplemental Information 1 Raw data for hormone levels, body weight, behavioural assays, and protein expression (Figures 1, 2, and 6).

Click here for additional data file.

Supplemental Information 2 Sequence Data.

Click here for additional data file.

Supplemental Information 3 Uncropped Gels/Blots.

Click here for additional data file.

Supplemental Information 4 Author Checklist.

Click here for additional data file.

Additional Information and Declarations

Competing Interests

Author Contributions

Animal Ethics

DNA Deposition

Data Availability

The authors declare that they have no competing interests.

Feng Zhao conceived and designed the experiments, performed the experiments, analyzed the data, prepared figures and/or tables, and approved the final draft.

Kai Wang performed the experiments, prepared figures and/or tables, and approved the final draft.

Yujun Wen analyzed the data, authored or reviewed drafts of the article, and approved the final draft.

Xiaohui Chen analyzed the data, authored or reviewed drafts of the article, and approved the final draft.

Hongya Liu analyzed the data, prepared figures and/or tables, and approved the final draft.

Faqiu Qi analyzed the data, authored or reviewed drafts of the article, and approved the final draft.

Youjuan Fu performed the experiments, prepared figures and/or tables, and approved the final draft.

Jiashu Zhu analyzed the data, prepared figures and/or tables, authored or reviewed drafts of the article, and approved the final draft.

Suzhen Guan conceived and designed the experiments, prepared figures and/or tables, and approved the final draft.

Zhihong Liu conceived and designed the experiments, analyzed the data, prepared figures and/or tables, and approved the final draft.

The following information was supplied relating to ethical approvals (i.e., approving body and any reference numbers):

Experimental operation complies with the relevant regulations of the ethics committee of the Animal Experimental Center of Ningxia Medical University (Number:IACUC-NYLAC-2019-089).

The following information was supplied regarding the deposition of DNA sequences:

The sequences are available at GenBank: PRJNA721070.

The following information was supplied regarding data availability:

The raw data is available in the Supplemental Files.

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
