# Peer review of "Contribution of hippocampal BDNF/CREB signaling pathway and gut microbiota to emotional behavior impairment induced by chronic unpredictable mild stress during pregnancy in rats offspring"

_PeerJ, doi:10.7717/peerj.13605_

## Round 0.1 · original submission · Major Revisions

Thank you for submitting the manuscript to PeerJ. It has been reviewed by experts in the field and we request that you make major revisions before it is processed further.

We look forward to hearing from you soon.

Best wishes,

Badicu Georgian, Ph.D

·

Basic reporting

General comments

Under my point of view, the English should be ameliorated and revised by a fluent English professional. The references must be updated, whereas old references must be removed.
The study was well designed and it shows innovative and relevant findings.
Some small changes must be made in order to ameliorate the manuscript, as described at the sequence.

Introduction Section
This section presents repeated sentences, whose focus involves the same issue, but in different words (lines 51-53 and 58-61). In this same section, the subject of the paper must be included, in any time the author explain about the microbiota, for example. In addition, conceptual itens must be included, but not discussion of the findings.

Results
The figure 1A and 1B involve experimental design and therefore must be separated from both figures 1C and 1D, which show results. In fact, these last figures must be presented in the Results section.

Experimental design

Experimental design:

The pivotal aim of the study must be clarified.
Authors presented a lot of text about stress, but little about the investigated BDNF/TrkB pathway and microbiota, which together are the differential point of the study.

Validity of the findings

The study has a good impact and is innovative. Underlying data has been provided; are robust, statistically sound and controlled. Conclusions are well-formulated, linked to the original research question, and limited to supporting results.

Additional comments

No more comments

Reviewer 2 ·

Basic reporting

Requires editing by fluent English writer. I suggest you have a colleague who is proficient in English and familiar with the subject matter review your manuscript, or contact a professional editing service.

The title should be improved. As a suggestion: “Chronic unpredictable mild stress during pregnancy impairs emotional behavior of rats offspring: contribution of hippocampal BDNF/CREB signaling and gut microbiota” or “Contribution of hippocampal BDNF/CREB signaling and gut microbiota to emotional behavior impairment induced by chronic unpredictable mild stress during pregnancy in rats offspring”.

Chronic unpredictable stress (CUMS) is a well-studied model. However, the study novelty should be clarifying. Please added a paragraph in the introduction section highlighting the freshness as well as the importance of the study.

Experimental design

The methodology should be clarified. A didactic form to better comprehension of the protocol is a schematic timeline indicating the days. Here are some examples in the literature (10.1007/s12035-020-02255-z, 10.1007/s12264-021-00754-0).

There are “Gender Differences in the Effects of Prenatal Stress on Brain Development and Behaviour – (s11064-007-9339-4)” and “Sex differences in depression: Insights from clinical and preclinical studies (10.1016/j.pneurobio.2019.01.006)”. In the present study, authors separated the animals offspring in each group of 16 rats (8 male rats,8 female rats). The results are expressed together, male and female mice. Please, discuss about the sex differences in the literature and assume this fragility as a limitation of the study.

The authors studied the effects of CUMS in maternal corticosterone, weight gain, movements and gut microbiota. However, the effects of CUMS in the hippocampus as well as the emotional behavior was accessed only in the rats offspring. If these effects were studied in the mother of the litter too, do the authors think there would be the same emotional behavior and hippocampal signaling of the pups?


The authors used (Li et al., 2020) as methodology for tail suspension test. The aforementioned study used mice rather than rats. In literature is suggested tail suspension test to mice and forced swimming test for mice and rats as methodology for depressive-like phenotype study. In agreement with the reviewer comment: “the forced swim (also termed behavioral despair) test in the rat and mouse, and the tail suspension test in the mouse (10.1002/0471142301.ns0810as55). Please, justify the rats use in the present study.

Validity of the findings

The “n” experimental should be specified in each technique. For example, some graphics contains data points which facilitates the comprehension. However, for example, in the Western blot technique, there is not the “n” specification as well as the data points. Please, pay attention.

Additional comments

In the discussion section line 465-466 “This study found that stress not only caused the maternal rat’s exercise capacity to be weakened”. Please, specify the parameter analyzed for these affirmation: “as demonstrated in horizontal movement”.

The protein CREB develops a critical role in pathogenesis of depression. Although the authors measured the CREB levels in the hippocampus of rats, there are a poorly discussion about this protein and its relevance for the study. Here are some suggestions of studies to improve the discussion section: (10.1016/j.biopsych.2005.11.003; 10.1002/bies.20658; 0.1016/j.bbr.2021.113245; j.neuropharm.2022.108990; 10.1016/j.bbr.2018.04.050).

Should the authors provide a mechanistic possible explanation of how CUMS is causing emotional behavior impairment. The reviewer encourages the authors to include a figure or graphical abstract for better understanding.

Annotated reviews are not available for download in order to protect the identity of reviewers who chose to remain anonymous.

Reviewer 3 ·

Basic reporting

All comments are below

Experimental design

All comments are below

Validity of the findings

All comments are below

Additional comments

The manuscript entitled “Chronic stress during pregnancy causes impairments in
emotional behavior of offspring may through BDNF/TrkB signaling pathway and gut
microbiota (#71239)” evaluated the impairments on behavioral and molecular parameters
in offspring from stresses mothers (rats). Furthermore, it was evaluated the gut microbiota
alteration and its relation with stress on previous generation. The author found alterations
on several parameters in mothers that show the validation of stress protocol. Some
changes in brain and gut microbiota were found in adult offspring from stressed mothers,
when compared to control group.

The results are consistent but some alterations (described below) could be made in order
to improve the quality of manuscript.

1. The whole manuscript must be revised by a fluent speaker;
2. Line 27: replace “gravid” by “pregnant”;
3. Line 28: replace the sentence “..and body weight and behavioural changes
recorded.” By “.The body weight was measured and the behavioral changes were
recorded.”
4. Line 32: replace “chronic stress” by CUMS.
5. Introduction: Paragraph 1 is too long, please divide it into 3 parts. I suggest divide
it at first in line 55 finishing with Plant et al., 2016. The second part I suggest divide
it in line 61, after Rotem-Kohavi et al., 2020.
6. Line 143: The term “canthal vein” is correct?
7. Line 151: I think rats don't received 1mL of blood, but 1mL was collected from
rats? If yes, please correct.8. The authors propose that stress can cause behavioral alteration through
BDNF/TrkB signaling pathway and gut microbiota. But the discussion lacks
information about relation between these two aspects (BDNF/TrkB signaling
pathway and gut microbiota). For example: The stressful events lead to a
modification in gut which lead to alteration in hippocampus or the modification
in hippocampus lead to gut alterations? Or there was no influence between these
two parameters? Please include some sentence about it
9. Discussion: Which consequences can happen when the gut microbiota is
modified, please discuss it.
10. Figure 6: In the figure of Western blotting (6d-i) authors just showed means and
SD. But SD does not appear in control group (black bar). Maybe authors could
change the color of bar, for SD to appear.
11. In all figures there were showed individual measurements (dots). In figure 6 could
appear the individual spots too.
12. Figure 6: There are only two experimental groups. For more confidence in results,
it could appear more than one representative band (maybe two).
13. Figure 6: TrkB band are not much representative with bars in figure 6e.

---

## Round 0.2 · accepted · Accept

Thank you for submitting the manuscript to PeerJ. Great improvements were performed in the manuscript. Currently, the article is acceptable for publication.

We look forward to hearing from you soon.

Best wishes,

Badicu Georgian, Ph.D

Reviewer 2 ·

Basic reporting

Clear and unambiguous, professional English used throughout.

Experimental design

Original primary research within Aims and Scope of the journal.

Validity of the findings

Conclusions are well stated, linked to original research question